# Transient inflammatory response mediated by interleukin-1β is required for proper regeneration in zebrafish fin fold

Tomoya Hasegawa[1], Christopher J Hall[2], Philip S Crosier[2], Gembu Abe[3], Koichi Kawakami[3], Akira Kudo[1], Atsushi Kawakami[1]*

[1]School of Bioscience and Biotechnology, Tokyo Institute of Technology, Midori-ku, Japan; [2]Department of Molecular Medicine and Pathology, School of Medical Sciences, University of Auckland, Auckland, New Zealand; [3]Division of Molecular and Developmental Biology, National Institute of Genetics, and Department of Genetics, SOKENDAI (The Graduate University for Advanced Studies), Mishima, Japan

**Abstract** Cellular responses to injury are crucial for complete tissue regeneration, but their underlying processes remain incompletely elucidated. We have previously reported that myeloid-defective zebrafish mutants display apoptosis of regenerative cells during fin fold regeneration. Here, we found that the apoptosis phenotype is induced by prolonged expression of *interleukin 1 beta* (*il1b*). Myeloid cells are considered to be the principal source of Il1b, but we show that epithelial cells express *il1b* in response to tissue injury and initiate the inflammatory response, and that its resolution by macrophages is necessary for survival of regenerative cells. We further show that Il1b plays an essential role in normal fin fold regeneration by regulating expression of regeneration-induced genes. Our study reveals that proper levels of Il1b signaling and tissue inflammation, which are tuned by macrophages, play a crucial role in tissue regeneration.

*For correspondence: atkawaka@bio.titech.ac.jp

**Competing interests:** The authors declare that no competing interests exist.

## Introduction

Progress in regenerative medicine depends on unraveling the mechanisms underlying tissue and organ regeneration. The zebrafish is a powerful model species for investigation of regeneration mechanisms because it exhibits high regenerative capacity. Zebrafish can regenerate complex structures such as fins, heart, brain, retina, and other tissues (*Gemberling et al., 2013*). The caudal fin of zebrafish, in particular, has been used as a model for analyzing epimorphic regeneration, a type of regeneration in the urodele limb and fish fin (*Poss et al., 2003*). During epimorphic regeneration, two characteristic tissues, the wound epidermis and the blastema, are induced in response to tissue amputation, and their coordinated actions regulate cell proliferation and morphogenesis and thus lead to tissue regeneration. Studies conducted using the caudal fin regeneration model have identified numerous genes and molecular signaling pathways that are critical for fin regeneration (*Yoshinari and Kawakami, 2011*; *Wehner and Weidinger, 2015*).

In addition to the classical regeneration model developed using the adult caudal fin, a zebrafish larval fin fold model has been developed (*Kawakami et al., 2004*; *Mateus et al., 2012*), allowing identification of additional genes and signaling pathways required for regeneration (*Mathew et al., 2009*; *Ishida et al., 2010*; *Yoo et al., 2012*). Specifically, the fin fold regeneration model has facilitated genetic analysis of the regeneration mechanism by exploiting a number of mutant zebrafish resources (*Rojas-Muñoz et al., 2009*; *Yoshinari et al., 2009*), because several

**eLife digest** Animals and other multicellular organisms all have at least some ability to regenerate lost or wounded tissues. Zebrafish are particularly good at this to the extent that they can replace damaged or lost body parts with exact replicas of the originals. In 2015, a team of researchers found that some mutant zebrafish that lack blood cells including immune cells are unable to regenerate lost tissues. This is because the cells that are primed to regenerate die instead, but it was not clear why this happens.

Many immune cells have roles in fighting infection and in responding to tissue damage.When a tissue is damaged, the area often becomes inflamed as white blood cells called macrophages flock to the damaged area to protect it from infection and remove damaged cells.

Hasegawa et al. – who include several researchers involved in the 2015 study – used genetic approaches to investigate the role of inflammation in tissue regeneration in zebrafish. The experiments show that several genes involved in inflammation – including one called *interleukin 1b* – were active over longer periods of time in the mutant fish compared with normal zebrafish. The gene produces a signal protein and this prolonged activity causes the primed regenerative cells to die. However, the cells can survive if *interleukin 1b* activity is quickly suppressed by macrophages. The experiments also show that, in order for tissues to regenerate properly, *interleukin 1b* needs to be active for only a short period of time.

The findings reveal that some inflammation is needed for tissues to regenerate, but that a more severe inflammatory response can block the process. A future challenge will be to identify the signals that macrophages produce to suppress inflammation to allow tissues to regenerate. These anti-inflammatory signals may have the potential to be used as drugs to cure chronic inflammatory diseases and boost tissue regeneration potential in humans.

zebrafish lethal mutants survive beyond 7 days post fertilization (dpf)—an adequate period for assaying tissue regeneration—because of nutrient supply from the yolk.

Previously, we reported that zebrafish mutants, *cloche* (*clo*) (**Stainier et al., 1995**; **Reischauer et al., 2016**) and *tal1/scl* (**Bussmann et al., 2007**), showed a unique regenerative defect: these mutants could not regenerate their fin fold because of apoptosis of the regenerative cells (**Hasegawa et al., 2015**). Our analyses of the *clo* mutant revealed that a diffusible survival factor from myeloid-lineage cells is required to prevent apoptosis of the primed regenerative cells. That study suggested that the regenerative cells are sensitive to apoptosis during tissue regeneration, but the mechanism of this sensitization and the identity of the survival factor remain unknown.

Myeloid cells are widely recognized to play a role in protection against pathogens and microorganisms after tissue injury. Neutrophils are first recruited to sites of inflammation, and these cells encounter pathogens and phagocytose and kill microorganisms by producing reactive oxygen species and/or antibacterial proteins (**Kolaczkowska and Kubes, 2013**). Subsequently, macrophages infiltrate the inflammation site, produce pro- or anti-inflammatory cytokines, remove tissue debris, secrete growth factors, and support tissue restoration (**Koh and DiPietro, 2011**).

Several recent studies have used the zebrafish model and addressed the roles of myeloid cells in response to tissue injury: Li and coworkers (**Li et al., 2012**) showed that knockdown of macrophage differentiation delayed fin fold regeneration and resulted in formation of an abnormal fin fold featuring large vacuoles. Furthermore, Petrie and coworkers (**Petrie et al., 2014**) showed that genetic ablation of macrophages in adult zebrafish impaired fin outgrowth and often induced an abnormal fin. These studies suggest a crucial role for macrophages in tissue regeneration; however, the mechanism by which myeloid cells affect regeneration remains to be elucidated.

Here, we show that a pro-inflammatory cytokine gene, *interleukin 1 beta* (*il1b*), is aberrantly upregulated in the *clo* mutant, and that the resulting excessive and prolonged inflammation causes apoptosis of the regenerative cells. Moreover, although myeloid cells are widely regarded as the main cells that secrete pro-inflammatory cytokines such as Il1b and thereby promote inflammation, we demonstrate that epidermal cells surrounding the amputated tissue are the source of Il1b in both wild-type (WT) and *clo* larvae. Notably, *il1b* expression in the regenerating epidermis is usually

quenched by the action of macrophages, and a lack of this anti-inflammatory effect of the macrophages results in apoptosis. Lastly, we show that Il1b signaling plays an indispensable role in normal tissue regeneration by activating expression of regeneration-induced genes. Thus, our study highlights the function of Il1b and the inflammatory response in tissue regeneration. A proper level of *il1b* expression and the inflammatory response, which are tuned by macrophages, are necessary for progression of tissue regeneration.

## Results

### Prolonged *il1b* expression in the *clo* mutant during regeneration

To understand the molecular pathway leading to apoptosis of regenerative cells in the *clo* mutant, we performed a transcriptome analysis of amputated larval tail tissues of WT and *clo* mutant zebrafish at approximately 6 hr post amputation (hpa), just before the stage at which apoptosis is first detectable in the *clo* mutant (*Hasegawa et al., 2015*). Compared with WT larvae, the *clo* larvae showed marked upregulation of regeneration-induced genes such as *junba*, *junbb*, *matrix metallopeptidase 9* (*mmp9*), *fibronectin 1b* (*fn1b*), and *fgf20a* (*Yoshinari et al., 2009*; *Shibata et al., 2016*) (*Figure 1A* and *Supplementary file 1*). In addition to these genes, expression of a group of genes including *il1b* and *tumor necrosis factor-b* (*tnfb*), which are involved in the inflammatory response, was also upregulated in the *clo* mutant during fin fold regeneration. Excluding enzymes and uncharacterized proteins, *il1b* was the most upregulated gene. *il1b* expression in the *clo* mutant was also confirmed using reverse transcription polymerase chain reaction (RT-PCR) analysis (*Figure 1B*). Notably, *prostaglandin2a* (*ptgs2a*), a downstream gene in Il1b signaling (*Dinarello, 2009*), was also upregulated in the *clo* mutant (*Figure 1A*), which clearly indicated that Il1b signaling was activated in the *clo* mutant.

Next, we performed in situ hybridization (ISH) analysis of *il1b* to reveal its spatiotemporal expression during regeneration. At 3 hpa, strong *il1b* expression was observed in both WT and *clo* mutant larvae (*Figure 1C and D*). In WT larvae, *il1b* was expressed in a group of scattered cells that appeared to be myeloid cells (*Figure 1C*; arrowheads) and also expressed at the distal edge of the fin fold. After 3 hpa, *il1b* expression was rapidly diminished in WT larvae, and most larvae displayed very weak or no *il1b* expression at 6 hpa. In contrast to the expression in WT, *il1b* expression in the *clo* mutant was maintained at 6 hpa and was still detectable at 12 hpa, indicating a prolonged inflammatory reaction in the *clo* mutant.

Bacterial infection is recognized to trigger an inflammatory response that accompanies the induction of *il1b* expression (*Dinarello, 2009*). To investigate whether the same mechanism underlies the induction of *il1b* expression by fin fold amputation and bacterial infection, we injected zebrafish with *Salmonella typhosa* lipopolysaccharide (LPS), a reagent whose injection mimics bacterial infection (*Lu et al., 2008*). WT larvae injected with LPS displayed systemic induction of *il1b* in cells that appeared to be myeloid cells (*Figure 1E*). By contrast, the *clo* mutant larvae injected with LPS showed few *il1b*-positive cells, which indicated that tissue injury and bacterial infection induce *il1b* expression through distinct mechanisms.

### *il1b* is expressed in epithelial cells during fin fold regeneration

The absence of myeloid cells in the *clo* mutant suggests that the *il1b* is expressed in non-myeloid cells. Accordingly, examination of tissue cryosections of the *clo* fin fold after *il1b* ISH analysis suggested that *il1b* was expressed in epidermal cells (*Figure 2A*).

To further investigate the identity of the *il1b*-expressing cells, we generated a transgenic (Tg) line, Tg(*il1b:egfp*), by introducing the *egfp-nitroreductase* fusion gene into the *il1b* locus of a bacterial artificial chromosome (BAC) clone, CH211-147H23. The larvae of Tg(*il1b:egfp*) showed faint EGFP expression in the entire epidermis and weak EGFP expression in the pectoral fins and at the end of the notochord (*Figure 2—figure supplement 1A*; arrowheads). After fin fold amputation, *egfp* expression was induced in a similar pattern to endogenous *il1b* expression (*Figure 2B*; *Figure 2—figure supplement 1B*), although EGFP fluorescence was detected at later stages of regeneration than *il1b* mRNA, probably as a result of the relatively longer half-life of the protein EGFP. LPS injection also induced EGFP fluorescence in myeloid cells (*Figure 2—figure supplement 1C*),

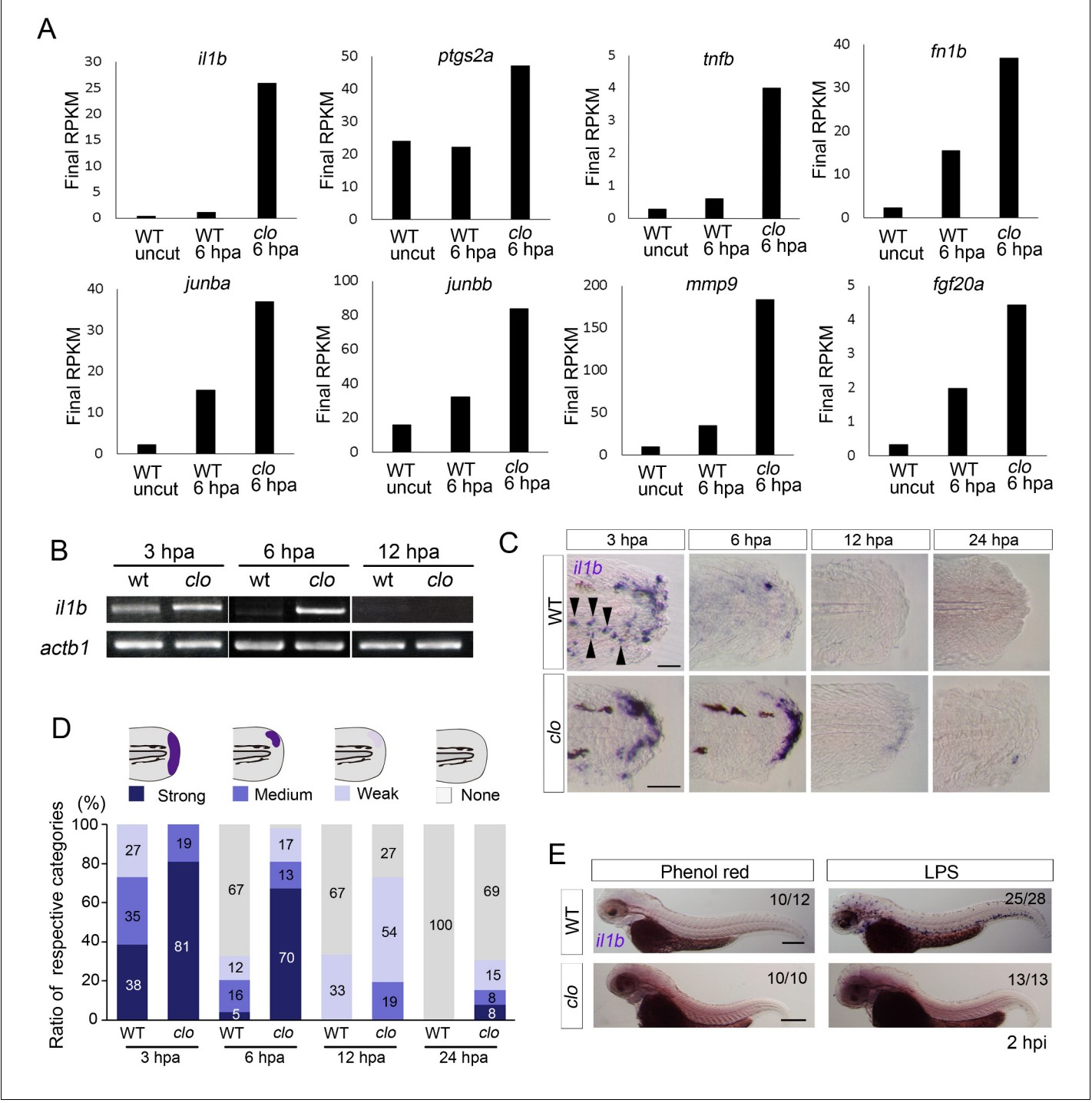

**Figure 1.** Prolonged activation of *il1b* expression in the *clo* mutant after fin fold amputation. (**A**) Transcriptome analysis of posterior fin fold tissues from uncut WT, amputated WT, and amputated *clo* mutant at 6 hpa. The ordinate, final RPKM, indicates the reads per kilobase of exon model per million mapped reads. Expression of regeneration-induced genes such as *fn1b*, *junba*, *junbb*, *mmp9*, and *fgf20a* was upregulated in the *clo* mutant. Notably, expression of inflammation-related genes such as *il1b*, *tnf1b*, and *ptgs2a* was highly upregulated in the amputated *clo* mutant. (**B**) RT-PCR analysis of *il1b* expression in WT and *clo* larvae at 3, 6, and 12 hpa. *actb1: actinb1*. (**C**) ISH analysis of *il1b* expression at corresponding stages of regeneration in WT and *clo*. Arrowheads indicate myeloid-like cells expressing *il1b*. Scale bars, 25 μm. (**D**) Quantification of *il1b* expression. The level of *il1b* expression was divided into four groups according to the ISH signal; the numbers in the bars indicate the ratios of the respective groups. Data are the sum of three experiments (total n > 13 for each sample). The data suggest that *il1b* expression extended into later stages in *clo* than in WT. (**E**) ISH analysis of *il1b* expression after LPS injection. Phenol red: negative control containing only phenol red; hpi: hours post injection. Scale bars, 200 μm.

*Figure 1 continued on next page*

*Figure 1 continued*

The following source data is available for figure 1:

**Source data 1.** *il1b* expression at respective time points after fin fold amputation.

which indicated that the *il1b:egfp* transgene recapitulated endogenous *il1b* expression in myeloid cells and also in the injured fin fold.

Next, using the Tg(*il1b:egfp*) reporter line, we confirmed that the EGFP-positive cells in the transgenic line mostly colocalized with the epithelial marker E-cadherin (*Figure 2C*), in both WT and the *clo* mutant; this finding indicated that *il1b* expression was principally induced in epithelial cells in response to tissue amputation. In addition to detecting *il1b* expression in the injured tissue, we detected *il1b* expression in WT larvae in migrating cells that are likely to be the myeloid cells (*Video 1*).

## Il1b signaling and inflammation are responsible for apoptosis in the *clo* mutant

Excessive inflammation has been suggested to directly or indirectly induce apoptosis (*Wallach et al., 2014*). Therefore, because elevated expression of the pro-inflammatory cytokine Il1b leads to chronic inflammation (*Dinarello, 2011*), we suspected that excessive *il1b* expression in the *clo* mutant could be a cause of the apoptosis of regenerative cells.

First, we used the Tg(*il1b:egfp*) reporter line to examine the relationship between *il1b*-expressing cells and apoptotic cells. Although many of the *il1b*-expressing cells did not overlap with the apoptotic cells that were mainly distributed in the mesenchyme (*Hasegawa et al., 2015*), the cells lay in close apposition (*Figure 3A*).

Next, to demonstrate the role of Il1b in the apoptosis occurring in the *clo* mutant, we knocked down *il1b* expression using an antisense morpholino oligonucleotide (MO). The *il1b* MO1 targeted to the splice sites (*Nguyen-Chi et al., 2014*) effectively inhibited *il1b* mRNA splicing (*Figure 3B*). Notably, *il1b* knockdown in the *clo* mutant resulted in a reduction in apoptosis of regenerative cells, whereas in the *clo* mutant injected with the standard control MO (std MO), no reduction in apoptosis was detected (*Figure 3C and D*). A similar anti-apoptosis effect of *il1b* knockdown was also obtained using another MO against *il1b* (*Yan et al., 2014*) (*Figure 3—figure supplement 1*).

To examine whether excessive inflammation is a cause of the apoptosis in the *clo* mutant, we used dexamethasone (Dex), a synthetic glucocorticoid that has been reported to suppress inflammation and *il1b* expression (*Kern et al., 1988*). Our results showed that Dex treatment abolished *il1b* expression in the *clo* mutant starting from an early stage of regeneration (*Figure 3E*), which indicated that the treatment effectively suppressed inflammation and *il1b* expression. In the Dex-treated *clo* larvae, the regenerative cells were rescued from apoptosis, much as in the *il1b*-knockdown larvae (*Figure 3F and G*). Collectively, these data support the notion that excessive Il1b signaling and inflammation in response to tissue injury represent a cause of apoptosis in the *clo* mutant.

## Macrophages support survival of regenerative cells

We have previously shown that the presence of myeloid cells is essential for survival of regenerative cells (*Hasegawa et al., 2015*). Myeloid cells such as neutrophils and macrophages have been shown to transiently migrate to the injured fin fold (*Li et al., 2012*). Actually, we observed that the myeloid cell accumulation in the injured fin fold reached a maximum between 3 and 6 hpa (*Figure 4A and B*). The timing coincides with the stage at which the initial *il1b* expression immediately after wounding was downregulated (*Figure 1C and D*). This suggests that myeloid cell migration could play a role in attenuating *il1b* expression, although we cannot conclude that myeloid cell migration is necessary for *il1b* downregulation.

Myeloid cells comprise several different cell types, including neutrophils and macrophages, but the cell type required for the survival of regenerative cells remains to be determined. Here, we used *csf3r*, *irf8*, and *spi1b* (*pu.1*) MOs, which inhibited the differentiation of neutrophils, macrophages, and both types of cells, respectively, and examined whether the depletion of neutrophils or

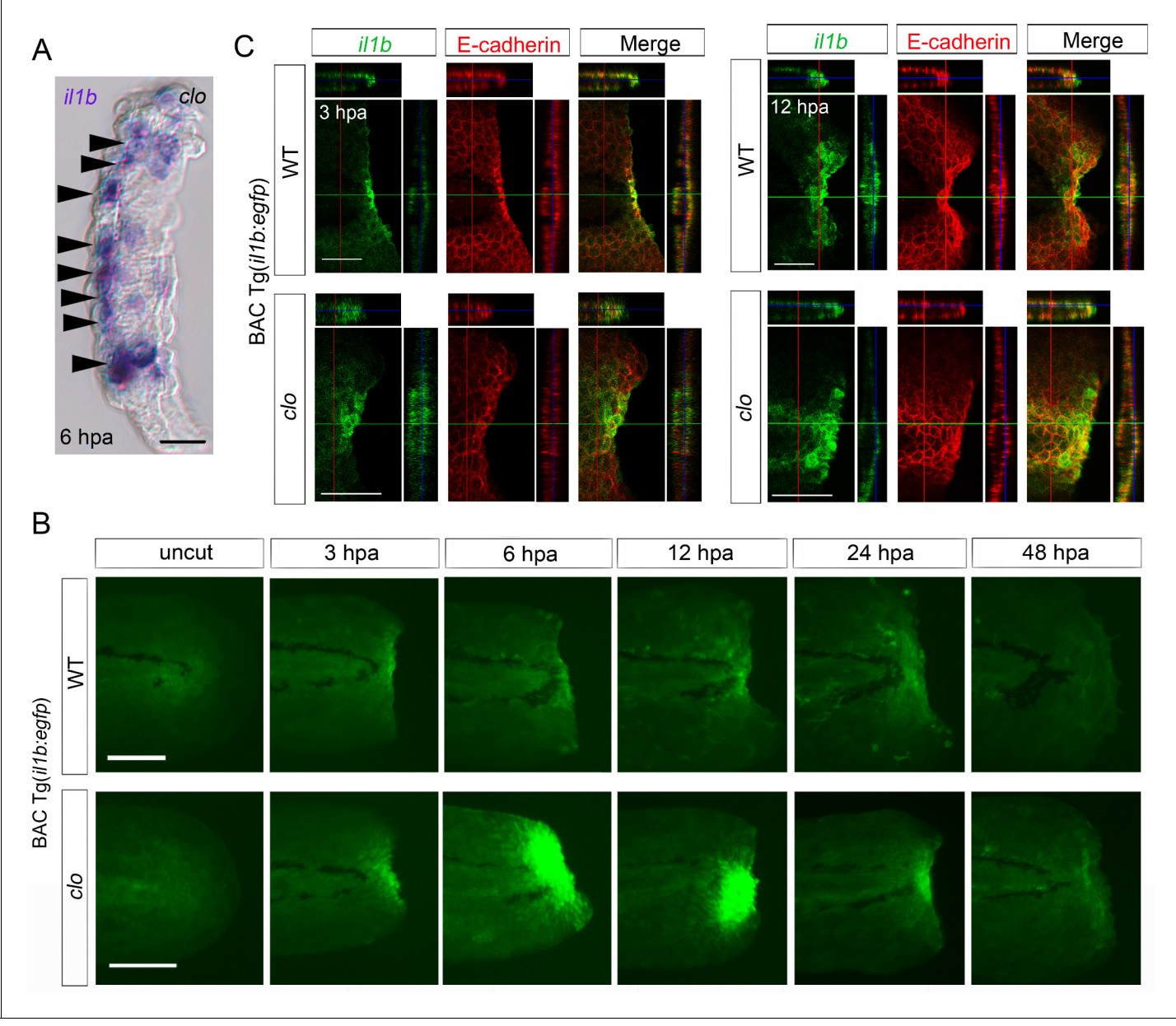

**Figure 2.** Spatiotemporal expression of *il1b* during fin fold regeneration. (A) Tissue cryosection of an amputated *clo* fin fold in which *il1b* expression was detected using ISH analysis. Arrowheads indicate the epidermal cells expressing *il1b*. Scale bar, 10 μm. (B) Live imaging of *il1b* expression in the Tg (*il1b:egfp*) line in WT and *clo* mutant at corresponding stages of regeneration. As in the case of *il1b* expression observed in ISH analysis (*Figure 1C*), EGFP fluorescence of the Tg(*il1b:EGFP*) was strongly upregulated in the *clo* mutant. The EGFP fluorescence in WT decreased more slowly than did the *il1b* mRNA signal, because of the comparatively longer half-life of the protein EGFP. Scale bars, 100 μm. (C) Immunofluorescence detection of EGFP and E-cadherin in Tg(*il1b:EGFP*) at 3 and 12 hpa. Confocal longitudinal and transverse optical sections are shown at the top and right side, respectively. Vertical and horizontal red and green lines indicate the approximate sites of the optical sections. Scale bars, 50 μm.

The following figure supplement is available for figure 2:

**Figure supplement 1.** Reproduction of *il1b* expression in the Tg(*il1b:egfp*) line.

macrophages induces apoptosis of regenerative cells as in the *clo* mutant. The efficacy of *csf3r* knockdown was confirmed by Sudan Black staining of neutrophils and ISH analysis using the *mpx* probe. In both the *csf3r* morphants and the *spi1b* morphants, fewer neutrophils were detected than in the control morphants (*Figure 4—figure supplement 1A,B,D*). In the *irf8* morphants, the

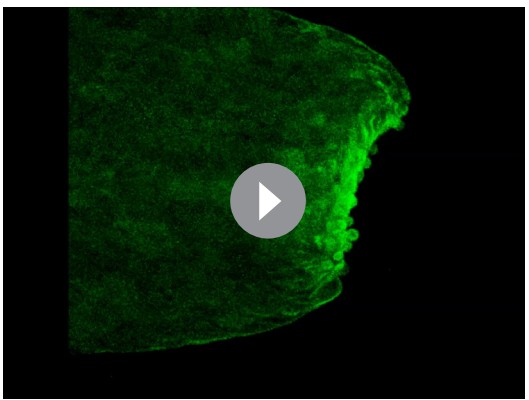

**Video 1.** Time-lapse imaging of EGFP fluorescence in Tg(*il1b:egfp*) from 1 to 11 hpa. Images were acquired once every 15 min. The epidermal cells at the injury site and the migrating cells, which are considered to be myeloid cells, expressed EGFP.

expression of *apoeb*, a marker for microglia—the resident macrophages in the central nervous system—was drastically decreased (*Figure 4—figure supplement 1C*). The decrease of macrophages was also evident in the wounded fin fold as revealed by *lcp1* ISH analysis (*Figure 4—figure supplement 1E*). None of these morphants showed any morphological abnormalities. These data indicated that the *irf8* MO effectively inhibited macrophage differentiation.

To examine apoptosis, we performed TUNEL staining in the aforementioned morphants after fin fold amputation, and the results showed that *irf8* morphants exhibited aberrant apoptosis of the regenerative cells, as did the *spi1b* morphants (*Hasegawa et al., 2015*) (*Figure 4C*). Importantly, similar numbers of TUNEL-positive cells were observed in the *irf8* and the *spi1b* morphants (*Figure 4D*). In sharp contrast to the *irf8* morphants, the *csf3r* morphants showed no increase in the apoptosis of regenerative cells. These results suggest that macrophages are responsible for supporting the survival of regenerative cells.

## Macrophages attenuate *il1b* expression and inflammation

We next investigated the mechanism by which macrophages support the survival of regenerative cells. First, we examined *il1b* expression at 6 hpa in the *spi1b*, *csf3r*, and *irf8* morphants. Whereas normal *il1b* expression was detected in the *csf3r* morphants, as in WT larvae, *il1b* expression remained elevated in the *spi1b* and *irf8* morphants, as in the *clo* mutant (*Figure 5A and B*). This result suggests that macrophages attenuate *il1b* expression and thereby prevent apoptosis of regenerative cells in WT larvae. Furthermore, the apoptosis induced in the *spi1b* or *irf8* morphants was rescued following *il1b* knockdown or Dex treatment (*Figure 5C–E*), which suggests that elevated *il1b* expression in the *spi1b* and *irf8* morphants is responsible for the observed apoptosis.

## *il1b* overexpression induces aberrant apoptosis and thereby suppresses fin fold regeneration

To further demonstrate that excessive *il1b* expression induces apoptosis, we generated and used the construct pTol2(*hsp70l:mCherry-T2a-il1b*), which we hereafter refer to as pTol2(*hsp70l:il1b*). The precise cleavage site of the zebrafish Il1b pro-peptide has not yet been identified (*Vojtech et al., 2012*), but based on comparing the zebrafish and human sequences, we estimated that the human IL-1$\beta$ cleavage site (D116) corresponds to the threonine residue at the 124th amino acid (a.a.) position in zebrafish Il1b (*Figure 6—figure supplement 1A*). Furthermore, to obtain constitutive secretion of this Il1b protein lacking a canonical signal-peptide sequence, the Il1b sequence after a.a. 125 was fused with the human IL-1$\beta$ receptor antagonist signal sequence (*Wingren et al., 1996*; *Tu et al., 2008*) and placed under the control of the promoter of *heat shock protein 70l* (*Figure 6A*). In the obtained Tg(*hsp70l:il1b*) line, mCherry fluorescence and *il1b* expression were observed after heat shock treatment (*Figure 6B*; *Figure 6—figure supplement 1B*), but not after stress caused by fin fold amputation (*Figure 6—figure supplement 1C*).

When TUNEL staining was performed after heat shock induction at 2 days post amputation (dpa), several TUNEL-positive cells were detected in the injured fin fold (*Figure 6C and D*), which indicated that the primed regenerative cells were susceptible to the excess Il1b signal. Furthermore, systemic and aberrant apoptosis was also observed when *il1b* was overexpressed in WT embryo from 12 to 24 hr post fertilization (hpf), at which stage myeloid cells have not fully differentiated (*Figure 6—figure supplement 1D*). This indicated that the excessive Il1b signaling potentially induces

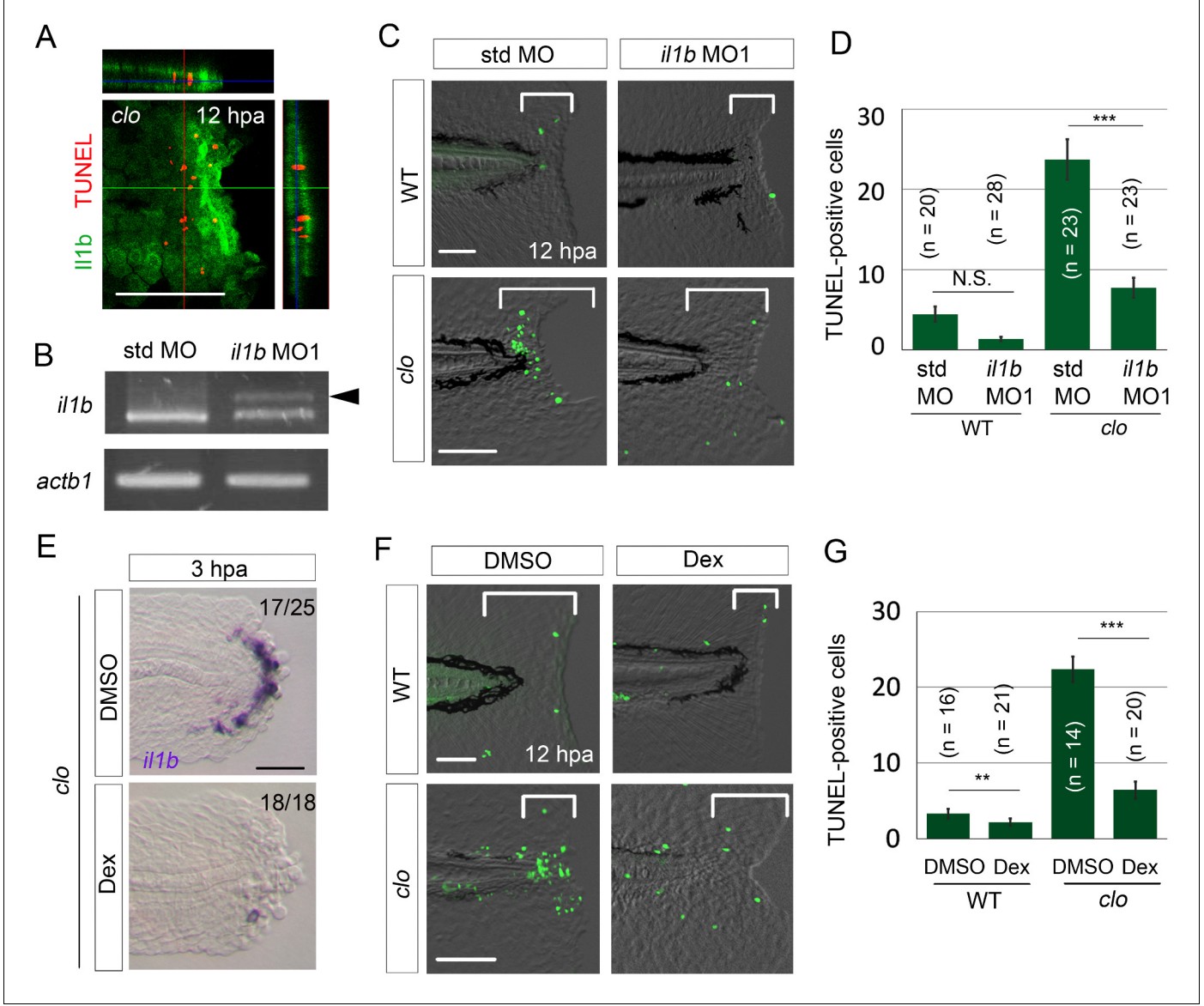

**Figure 3.** *il1b* knockdown or Dex treatment rescued the *clo* from apoptosis. (**A**) Simultaneous detection of apoptosis by TdT-mediated dUTP nick end labeling (TUNEL) and the *il1b*-expressing cell at 12 hpa in the *clo* mutant carrying the transgene *il1b:egfp*. Most of the TUNEL-positive cells were detected in close association with the *il1b*-expressing epithelial cells. Vertical and horizontal lines indicate the approximate sites of the optical sections. Scale bar, 100 µm. (**B**) RT-PCR analysis of *il1b* and *actinb1* expression in zebrafish larvae injected with *il1b* MO. The *il1b* MO was targeted to the splice donor site. Arrowhead indicates the aberrant transcript. (**C**) TUNEL analysis in amputated WT or *clo* larvae (12 hpa) after *il1b* knockdown. Injection of *il1b* MO substantially reduced the apoptosis of regenerative cells in the *clo* mutant, whereas the apoptosis was unaffected by std MO. Scale bars, 50 µm. (**D**) Quantification of TUNEL staining in the areas bracketed in (**C**). (**E**) ISH analysis of *il1b* expression at 3 hpa in larvae treated with Dex. DMSO: dimethyl sulfoxide, used as the vehicle. Dex treatment abolished *il1b* expression. Scale bar, 50 µm. (**F**) TUNEL staining in the amputated fin fold at 12 hpa in Dex-treated WT and *clo* larvae. Scale bars, 50 µm. (**G**) Quantification of TUNEL staining in the areas bracketed in (**F**). In (**D**) and (**G**), data are presented as means ± SEM. Student's *t* test, **p<0.01; ***p<0.001; N.S., not significant (p=0.172).

The following figure supplement is available for figure 3:

**Figure supplement 1.** Rescue from apoptosis by *il1b* MO2.

apoptosis not only in fin fold but also in other parts of the body. In addition to inducing the aberrant apoptosis, *il1b* overexpression suppressed the extension of the fin fold (*Figure 6E and F*).

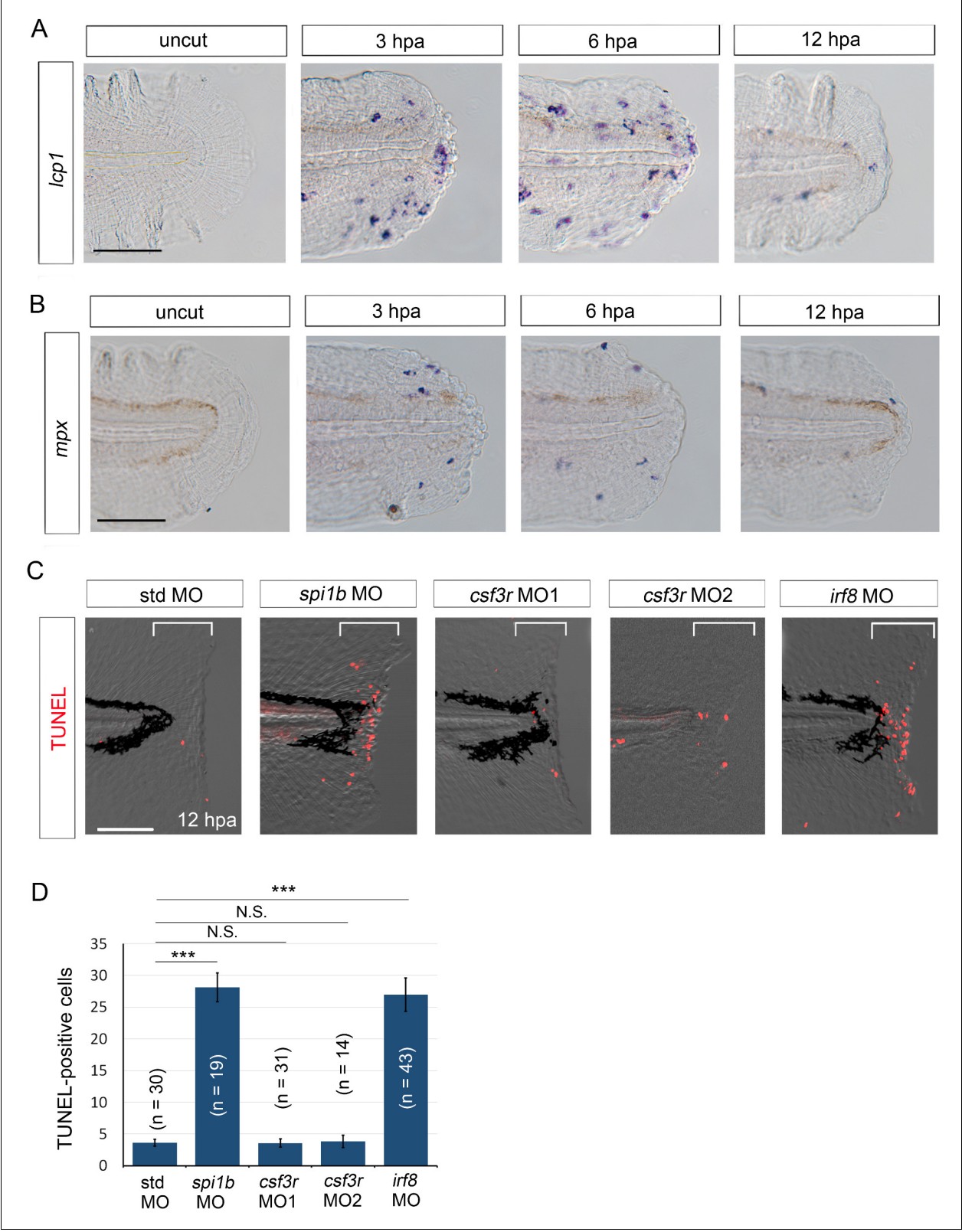

**Figure 4.** Macrophages are responsible for survival of regenerative cells. (**A**) ISH analysis of *lcp1* expression, a macrophage marker, during fin fold regeneration in WT. Scale bar, 50 μm. (**B**) ISH analysis of *mpx* expression, a neutrophil marker, during fin fold regeneration in WT. Scale bar, 50 μm. (**C**) TUNEL analysis in the amputated fin fold at 12 hpa after injection of *spi1b*, *csf3r*, and *irf8* MOs. Injection of *spi1b* MO and *irf8* MO markedly increased

*Figure 4 continued on next page*

*Figure 4 continued*
apoptosis of regenerative cells. Scale bar, 100 μm. (D) Quantification of TUNEL-positive cells in the areas bracketed in (A). In (B), data are presented as means ± SEM. Student's *t* test, ***p<0.001; N.S., not significant (p=0.981).
The following figure supplement is available for figure 4:

**Figure supplement 1.** Efficacy of MO-mediated macrophage and/or neutrophil depletion.

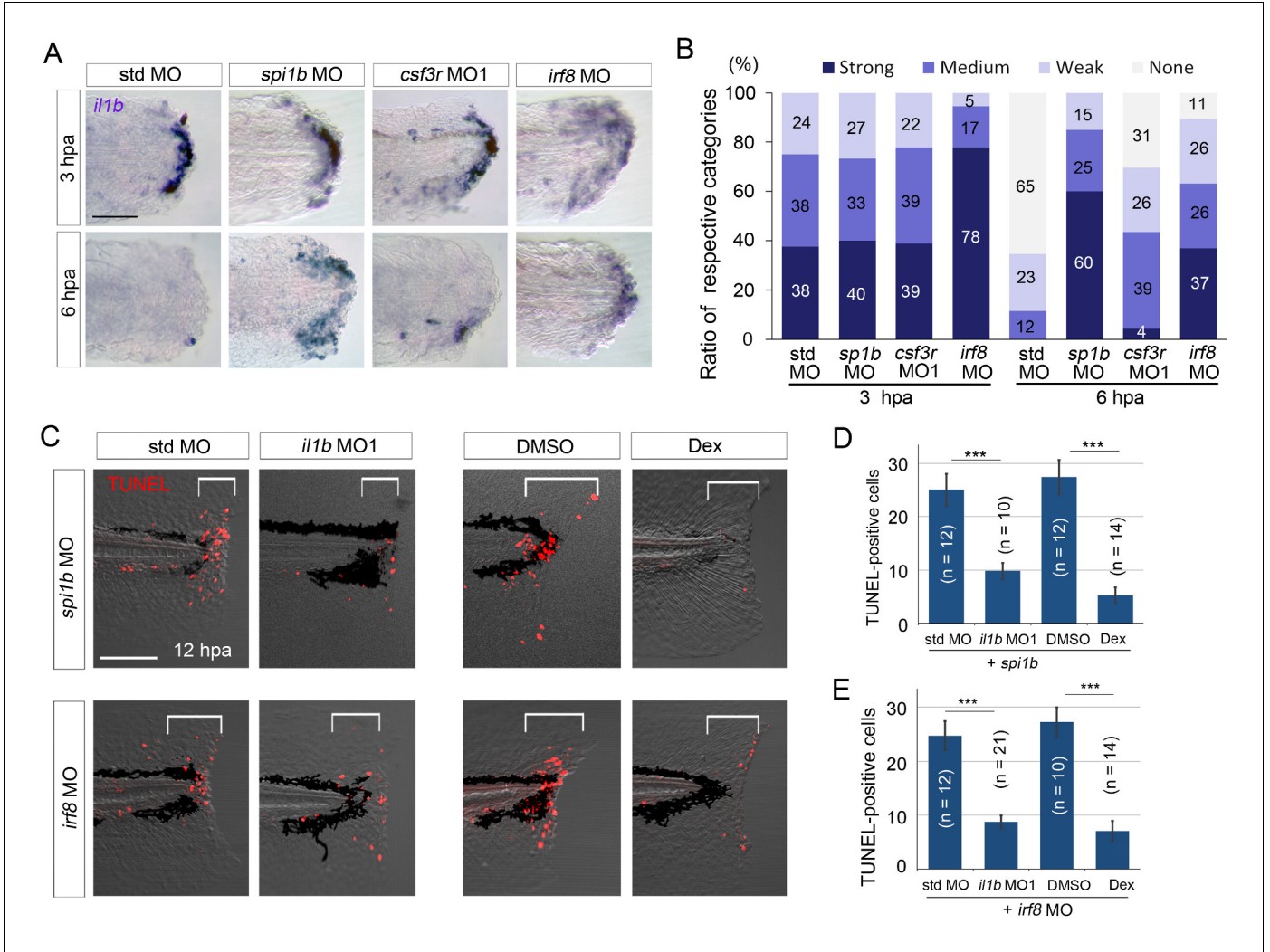

**Figure 5.** Prolonged *il1b* expression and apoptosis are induced by macrophage loss during fin fold regeneration. (A) ISH analysis of *il1b* expression in *spi1b*, *csf3r*, and *irf8* morphants. The *spi1b* and *irf8* morphants displayed prolonged *il1b* expression at 6 hpa. Scale bar, 50 μm. (B) Quantification of *il1b* expression detected using ISH analysis in (A). The level of *il1b* expression was evaluated as in *Figure 1D*. Data are the sum of two experiments (total n > 16 for each MO). (C) TUNEL analysis of the amputated fin fold at 12 hpa after *il1b* knockdown or Dex treatment in larvae in which macrophages were depleted using the *spi1b* or *irf8* MO. Scale bar, 100 μm. Apoptosis caused by *spi1b* and *irf8* MOs was rescued by *il1b* MO administration or Dex treatment. (D) Quantification of TUNEL staining in the *spi1b* morphants (C; bracketed areas). (E) Quantification of TUNEL staining in the *irf8* morphants in (C). In (D) and (E), data are presented as means ± SEM. Student's *t* test, ***p<0.001.
The following source data is available for figure 5:

**Source data 1.** *il1b* expression in larvae injected with std, *spi1b*, *csf3r*, or *irf8* MOs.

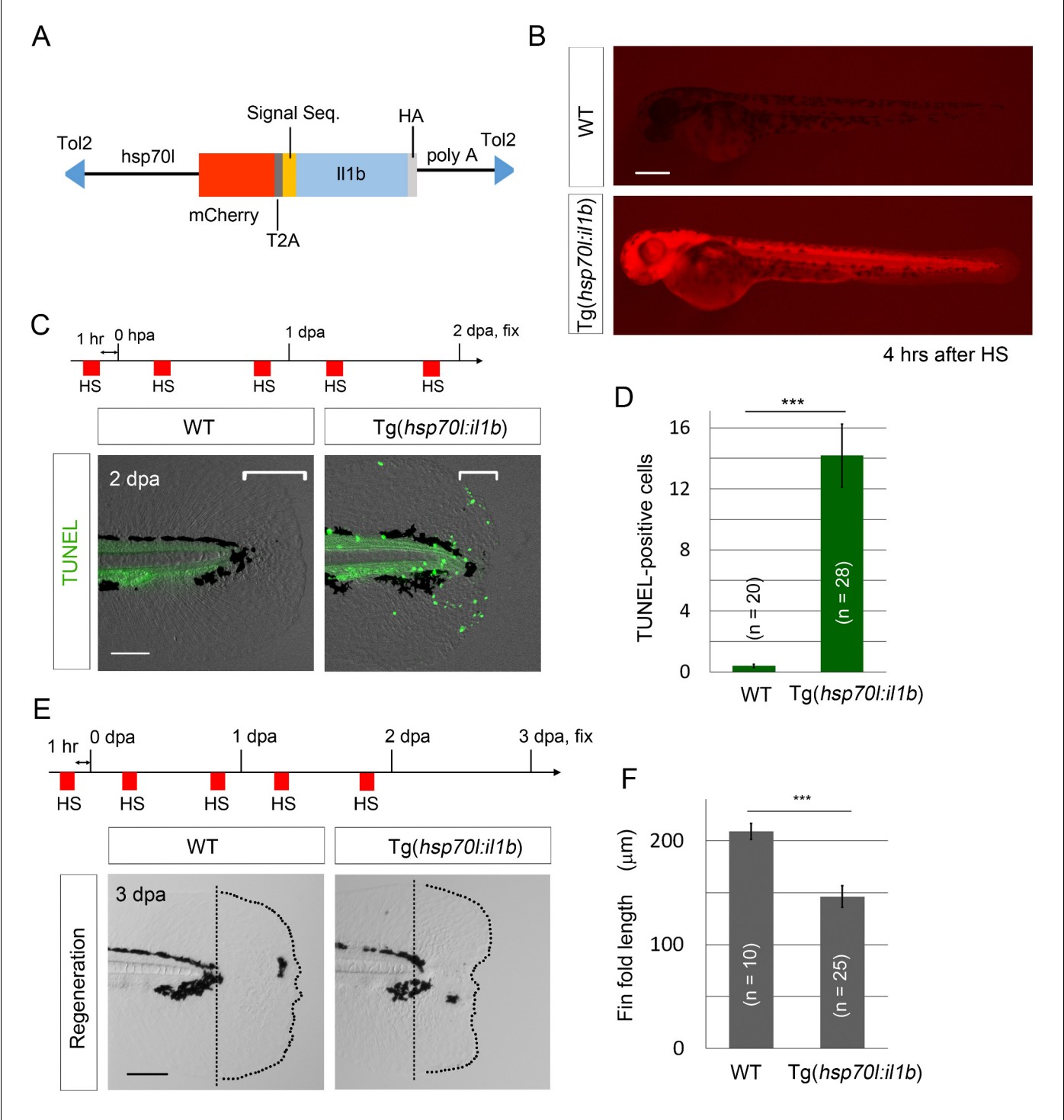

**Figure 6.** *il1b* overexpression induces aberrant apoptosis. (**A**) A schematic diagram of the construct used; in this construct, the zebrafish *heat shock protein 70l* promoter drives the expression of *mCherry* and *il1b* harboring a signal sequence. (**B**) mCherry fluorescence in Tg(*hsp70l:il1b*) at 4 hr after heat shock. WT, sibling WT. Scale bar, 250 μm. (**C**) TUNEL analysis in the amputated fin fold of Tg(*hsp70l:il1b*) at 2 dpa. The diagram depicts the experimental procedure. HS, heat shock. Scale bar, 100 μm. (**D**) Quantification of TUNEL staining in (**C**). (**E**) Fin fold regeneration in Tg(*hsp70l:il1b*) and sibling WT larvae. Dotted lines indicate amputation planes and fin fold outlines. Scale bar, 100 μm. (**F**) Quantification of the length of the regenerate (length posterior to the amputation plane; shown in (**E**)). In (**D**) and (**F**), data are presented as means ± SEM. Student's *t* test, ***p<0.001.

The following figure supplement is available for figure 6:

*Figure 6 continued on next page*

*Figure 6 continued*

**Figure supplement 1.** Generation of Tg(*hsp:il1b*).

## Normal inflammation mediated by Il1b is required for fin fold regeneration

Our data thus far suggest that excessive Il1b-mediated inflammation negatively affects the survival of regenerative cells and tissue regeneration in the *clo* mutant. However, inflammation has also been suggested to be essential for complete tissue regeneration (*Mathew et al., 2007*; *King et al., 2012*; *Kyritsis et al., 2012*). To examine the role of tissue inflammation and Il1b during normal fin fold regeneration, we used Dex and *il1b* MO1 and tested whether the inflammation mediated by Il1b affects normal fin fold regeneration.

To assess the role of inflammation in tissue regeneration, we administered Dex to WT larvae and assessed fin fold regeneration. The Dex-treated larvae displayed apparent retardation of regeneration at 3 dpa (*Figure 7A and B*) and a statistically significant reduction in cell proliferation in the distal fin fold region (*Figure 7C and D*). Notably, the expression of regeneration-induced genes such as *junba* (*Yoshinari et al., 2009*) and *fgf20a* (*Whitehead et al., 2005*; *Shibata et al., 2016*) was downregulated in the Dex-treated larvae (*Figure 7E*). These data suggest that inflammation is required for activation of regeneration-induced gene expression which is a prerequisite for normal regeneration.

Next, we used *il1b* knockdown in WT zebrafish to demonstrate the role of Il1b signaling in normal regeneration: *il1b* knockdown induced a similar phenotype to Dex treatment (*Figure 7F–I*), although the phenotype was slightly milder than that of the Dex-treated larvae, probably because of lower penetrance of the MO knockdown. Importantly, similar to the Dex-treated larvae, the *il1b* morphants showed attenuated expression of *junba* and *fgf20a* (*Figure 7J*). Collectively, these results suggest that the inflammatory reaction mediated by Il1b plays a necessary role in normal fin fold regeneration.

Lastly, we tested whether *il1b* overexpression induces a marked increase in expression of the regeneration-induced genes *junba*, *junbb,* and *fn1b* in uninjured tissue (*Figure 7K*). As expected, *il1b* overexpression induced the ectopic expression of *fn1b*, *junba*, and *junbb,* both in the uncut fin fold (*Figure 7L*) and in several other tissues such as the pectoral fins. Importantly, the apoptosis induced by *il1b* overexpression was only detectable at 2 days after multiple heat shocks (*Figure 6C and D*), possibly because of the anti-inflammatory function of macrophages in WT. But, induction of *junba*, *junbb,* and *fn1b* was observed at 12 hr after two heat shocks, a stage at which the apoptosis induced by *il1b* overexpression has not taken place (*Figure 7—figure supplement 1*), indicating that expression of the regeneration-induced genes was not caused by an indirect regenerative response to cell death. These data indicate that Il1b signaling is required tostimulate expression of regeneration-induced genes and regulate the initiation of fin fold regeneration.

## Discussion

It has been suggested that myeloid cells play crucial roles in the inflammatory responses of injured tissues. The pro-inflammatory cytokines (such as Il1b) produced by myeloid cells, including macrophages, induce early responses against infection or injury (*Dinarello, 2009*). Thus, myeloid cells are considered to trigger inflammation by providing pro-inflammatory cytokines. Here, we demonstrated that the pro-inflammatory cytokine Il1b is provided by epidermal cells in response to tissue injury. We showed that apoptosis occurred if the Il1b action at the injury site was prolonged, and that macrophages were responsible for attenuation of *il1b* expression and resolution of inflammation. Furthermore, our data suggest that normal *il1b* expression and inflammatory response are necessary in tissue regeneration to activate expression of regeneration-induced genes. Thus, our study has revealed that Il1b signaling and tissue inflammation act as a double-edged sword: they are required for regeneration, but in excess, they impair tissue regeneration (*Figure 8*).

Myeloid cells are considered to be the principal producers of Il1b, although certain studies have reported *il1b* expression in melanoma cells (*Okamoto et al., 2010*) and human keratinocytes

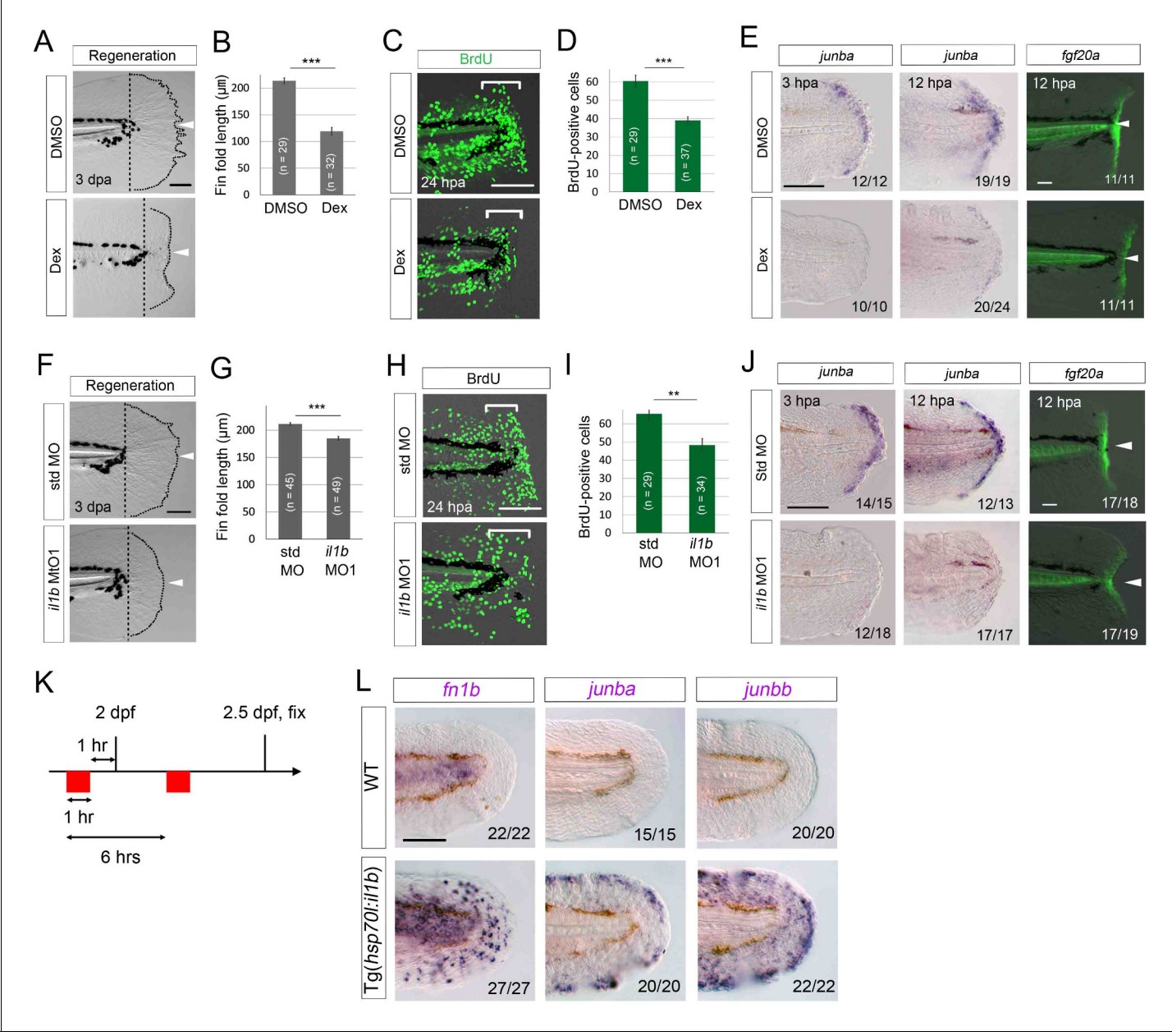

**Figure 7.** Il1b-mediated inflammation is required for normal regeneration. (A) Fin fold regeneration in WT larvae treated with Dex. Regeneration was retarded following treatment with Dex but not vehicle DMSO (arrowheads). Scale bar, 100 µm. (B) Quantification for (A) (posterior to the notochord). (C) BrdU incorporation in larvae treated with DMSO or Dex. BrdU labeling, 0–24 hpa. Scale bar, 50 µm. (D) Quantification for (C) (bracketed areas). BrdU-positive cells were significantly reduced following Dex treatment. (E) Detection of *junba* expression using ISH and *fgf20a* expression using the *HGn21A* enhancer-trap line (*Shibata et al., 2016*) in DMSO- or Dex-treated larvae. The expression of *junba* and *fgf20a* was also significantly downregulated following Dex treatment. Arrowheads indicate EGFP expression at amputation sites. Scale bars, 50 µm (F) Fin fold regeneration in larvae injected with std or *il1b* MO. The knockdown of *il1b* reduced fin fold regeneration. Scale bar, 100 µm. (G) Quantification for (F) (posterior to the notochord). (H) BrdU incorporation in larvae injected with std or *il1b* MO. BrdU labeling, 0–24 hpa. Scale bar, 50 µm. (I) Quantification for (H) (bracketed areas). (J) Detection of *junba* expression using ISH and *fgf20a* expression using the *HGn21A* line in larvae injected with std or *il1b* MO. The expression of *junba* and *fgf20a* was downregulated following *il1b* knockdown as in the Dex-treated larvae. Arrowheads indicate EGFP expression atamputation sites. Scale bars, 50 µm. (K) Schematic of the procedure of *il1b* overexpression in uninjured WT larvae. Heat shock was applied twice before fixation. (L) ISH analysis of *fn1b*, *junba*, and *junbb* in WT and Tg(*hsp70l:il1b*). *il1b* overexpression stimulated ectopic expression of regeneration-induced genes in uninjured larvae. Scale bar, 50 µm. In (A) and (F), dotted lines indicate amputation planes and fin fold outlines. In (B), (D), (G), and (I), data are presented as means ± SEM. Student's *t* test, **p<0.01; ***p<0.001.

*Figure 7 continued on next page*

*Figure 7 continued*

The following figure supplement is available for figure 7:

**Figure supplement 1.** Absence of apoptosis after two HS inductions in the Tg(*hsp70l:il1b*).

(*Kupper et al., 1986*). In this study, we showed that *il1b* is expressed by epidermal cells around the injury site. Notably, LPS injection induced *il1b* expression in myeloid cells, but not in the injured epidermal cells, suggesting that *il1b* induction by tissue injury occurs through a mechanism distinct from the mechanism underlying induction in myeloid cells following bacterial infection. Although it remains unknown how tissue injury stimulates *il1b* expression, a subject for future study, a substance released from disrupted cells (*Rock and Kono, 2008*) or a mechanical signal could be responsible for inducing *il1b* expression (*Kanjanamekanant et al., 2013*).

We also showed that in the *clo* mutant apoptosis is caused by prolonged *il1b* expression and inflammatory response. The precise mechanism by which Il1b induces apoptosis is unclear, but Tnf signaling could be a downstream mediator for triggering apoptosis. Tnfα signaling is known to stimulate apoptosis through activation of Caspase 8 (*Sedger and McDermott, 2014*). Tissue inflammation has been suggested to induce *tnfα* expression, and zebrafish *tnfb*, one of the zebrafish homologs of *tnfα*, was upregulated in the *clo* mutant (*Figure 1A*); therefore, Tnfα signaling potentially serves as a direct mediator for inducing apoptosis in the *clo* mutant. Alternatively, the *clo* apoptosis might occur because of ER stress signaling: Il1b administration into primary rat *β*-cells and

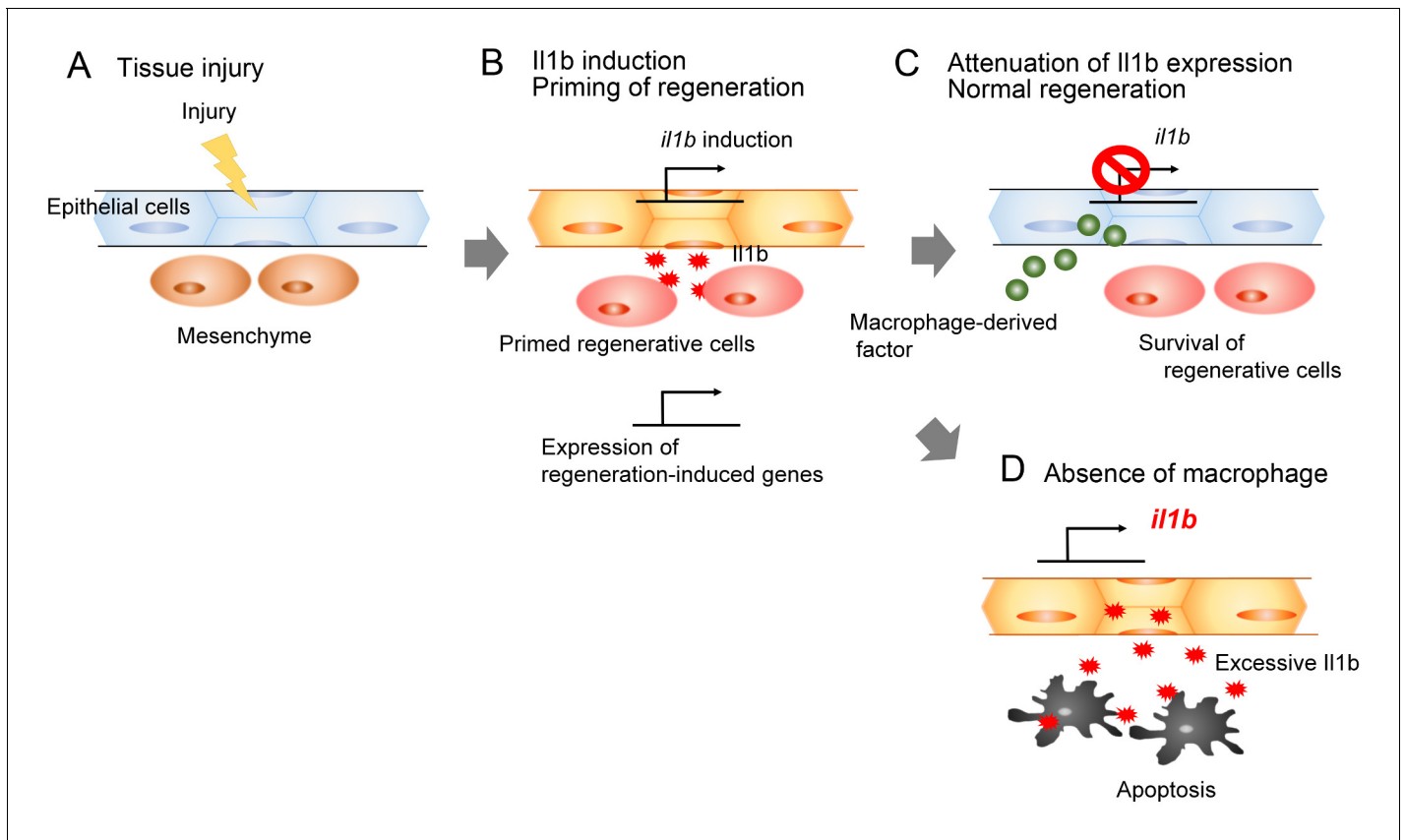

**Figure 8.** Schematic of the role of Il1b and macrophages during fin fold regeneration. (A and B) Tissue injury induces *il1b* expression in epithelial cells. The Il1b acts on the surrounding cells to prime them for regeneration by triggering expression of regeneration-induced genes such as *fn1b*, *junba*, *junbb*, and *fgf20a*. (C) The macrophage-derived factor subsequently attenuates *il1b* expression during normal regeneration. (D) In the absence of macrophages, apoptosis is induced in the primed regenerative cells. Thus, the Il1b function is a double-edged sword in tissue regeneration.

MIN6 cells was reported to increase ER stress through intracellular release of $Ca^{2+}$ and to induce apoptosis in a JNK-dependent manner (*Eizirik and Mandrup-Poulsen, 2001*; *Wang et al., 2009*). Thus, excessive ER stress evoked by Il1b around the injury site could induce apoptosis in the *clo* mutant.

In this study, we further demonstrated that macrophages play a crucial role in fin fold regeneration by attenuating *il1b* expression and supporting survival of regenerative cells. Recent studies have shown that knockdown of macrophage differentiation delayed fin fold regeneration and resulted in an abnormal fin fold featuring large vacuoles (*Li et al., 2012*), and genetic ablation of macrophages in adult zebrafish was demonstrated to affect fin ray patterning and fin growth (*Petrie et al., 2014*). These studies suggest that macrophages play a critical role in tissue regeneration, but the precise function of the cells remains unknown. Our study has, for the first time, elucidated a role of macrophages in attenuating *il1b* expression and inflammation during tissue regeneration. In accordance with our observation, macrophage-depleted mice and salamanders were found to display elevated *il1b* expression (*Goren et al., 2009*; *Godwin et al., 2013*), which suggests that the macrophage function of attenuating *il1b* transcription is conserved in other vertebrate species.

Our observation of macrophage accumulation in the injured fin fold suggests that the macrophage migration could play a role in attenuating *il1b* expression and quenching inflammation. However, we cannot conclude that the macrophage migration is necessary for attenuating the *il1b* expression, because our preceding study showed that an activity that rescues the *clo* apoptosis exists in the WT body extract prior to tissue amputation. We previously suggested that the effect of myeloid cells is mediated by a factor that is diffusible and heat-stable (*Hasegawa et al., 2015*), but its identity remains unknown. The factor could be an anti-inflammatory protein/molecule such as IL-10, Tgf-β, or a lipid mediator (*Serhan et al., 2008*; *Ortega-Gómez et al., 2013*). It is speculated that signals such as nFkB and/or STAT3, which are activated by tissue injury, are involved in inducing *il1b* expression (*Fang et al., 2013*; *Ogryzko et al., 2014*; *Karra et al., 2015*). The anti-inflammatory molecules bind to cell surface receptors to activate intracellular pathways and antagonize the nFkB and/or STAT3 signals to attenuate *il1b* transcription (*Yu et al., 2009*; *Saraiva and O'Garra, 2010*; *Liao et al., 2012*).

Most importantly, our study suggests that inflammation mediated by Il1b is also necessary for normal fin regeneration. Previously, we showed that genes such as *fn1b*, *junba*, and *junbb* were induced in response to tissue injury (*Yoshinari et al., 2009*), but the upstream mechanism that regulates the expression of these genes remains unknown. Here, we have demonstrated that Il1b signaling is necessary and sufficient for triggering expression of regeneration-induced genes. Although the primary factors involved in tissue regeneration, tentatively named alarmins (*Bianchi, 2007*), remain to be identified, Il1b could represent one of the key upstream mediators for initiating expression of regeneration-induced genes and advancing tissue regeneration.

Given that macrophages are required for resolving inflammation, an impairment or reduction of this function will lead to the diseases that accompany chronic inflammation, such as rheumatoid arthritis, inflammatory bowel disease, and auto-inflammatory diseases. Therefore, elucidation of the regulatory mechanism of the Il1b signal and the action of anti-inflammatory substances released from macrophages will enhance our understanding of the pathogenic processes of chronic-inflammation diseases and lead to development of suitable therapeutics.

## Materials and methods

### Fish husbandry and fin amputation

Zebrafish of a WT line, which has been maintained in our facility for >10 years through inbreeding, were housed in a recirculating system in a 14 hr day/10 hr night cycle at 28.5°C. When necessary, larvae were incubated in egg water (0.06% artificial marine salt, 0.0002% methylene blue) containing 0.003% phenylthiourea to prevent pigment formation. The zebrafish mutant strain *clo*[m39] (RRID: ZFIN_ZDB-ALT-980203-381), the *fgf20a* enhancer-trap line HGn21A (*Shibata et al., 2016*), the BAC Tg(*il1b:egfp*) line, and Tg(*hsp70l:il1b*) were used. The *clo*[m39] mutant was genotyped as previously described (*Hasegawa et al., 2015*). Zebrafish were subject to heat-shock induction at 38°C for 1 hr in a small water bath and then cooled to 28.5°C over time. Fin fold fin amputation was performed as previously described (*Kawakami et al., 2004*). Briefly, zebrafish larvae at 2 dpf were anesthetized

with 0.04% 3-amino benzoic acid ethylester (tricaine) in egg water and their fin fold was amputated using a surgical razor blade. For reproducible quantification, the fin fold was carefully amputated at sites just posterior to the notochords. For quantification of fin fold regeneration in *Figure 6F*, *Figure 7B and G*, the length from the notochord end to the posterior tip of fin fold was measured.

## RNA sequencing analysis

Larval posterior tissues (~1 mm from the distal end, approximately the tissue posterior to the yolk extension) from uncut WT, amputated WT, and amputated *clo* mutant, which were obtained from more than five independent batches of incrosses of *clo* heterozygotes, were collected on dry ice and stored at −80°C. Total RNAs from approximately 500 posterior tissue samples were extracted using TRIzol (Thermo Fisher Scientific, Waltham, MA) according to the manufacturer's instructions. RNA sequencing analysis was performed by Dragon Genomics (Kyoto, Japan). Briefly, purified and fragmented PolyA$^+$ RNAs were prepared from 3 μg each of the total RNAs using the TruSeq RNA sample preparation kit (Illumina, San Diego, CA), and the respective cDNAs were synthesized according to the manufacturer's instructions. The cDNAs fused with an adaptor on both ends were PCR-amplified (15 cycles) and used for sequencing analysis (Illumina HiSeq2000). The obtained sequence reads (60927373 reads for WT uncut, 59284653 reads for amputated WT at 6 hpa, 58507894 reads for amputated *clo* mutant at 6 hpa) were aligned to the zebrafish reference sequence using Bowtie software (version 0.12.7; RRID:SCR_005476) (*Langmead et al., 2009*). The normalized expression level of the respective genes, RPKM (reads per kilobase of exon model per million mapped reads), was calculated using ERANGE software (version 3.2; RRID:SCR_005240) (*Mortazavi et al., 2008*).

## MO injection

MOs (Gene Tools, Philomath, OR) were dissolved in Danieau solution. Fertilized zebrafish eggs were dechorionated by incubating them with 2% pronase (Roche) and then microinjected at the 1–2 cell stages. The following MOs were used in this study:
*irf8* MO: 5'-AATGTTTCGCTTACTTTGAAAATGG-3' (*Hall et al., 2014*)
*csf3r* MO1: 5'-ATTCAAGCACATACTCAC-TTCCATT-3' (*Hall et al., 2014*)
*csf3r* MO2: 5'- GAACTGGCGGATCTGTAAAGACAAA −3' (*Halloum et al., 2016*)
*spi1b* MO: 5'-GATATACTGATACTCCATTGGTGGT-3' (*Rhodes et al., 2005*)
*il1b* MO1: 5'-CCCACAAACTGCA-AAATATCAGCTT-3' (*Nguyen-Chi et al., 2014*)
*il1b* MO2: 5'- AAACGTAAAATAACTCACCATTGCA −3' (*Yan et al., 2014*)
std MO: 5'-CCTCTTACCTCAGTTACAATTTATA-3'

The *csf3r* MO2 (0.25 mM) and the rest of MOs (1 mM) were injected at approximately 0.5 nl per egg. Under these conditions, no apparent side effect was observed. Injected embryos were incubated in 0.3× Niu-Twitty solution.

## RT-PCR analysis

Total RNAs were extracted from larval tail tissues (200 for each) posterior to the yolk extension using TRIzol and further purified using the RNeasy kit (Qiagen, Venlo, Netherlands). Among the purified total RNAs, aliquots that corresponded to 90 larval tails were used for cDNA synthesis. The cDNAs were synthesized using the Thermoscript RT-PCR kit (Thermo Fisher Scientific) with random hexamers as the primer. The synthesized cDNAs were diluted to 0.1 μg/μl and stored at −30°C. PCR was performed according to a standard procedure using Paq5000 DNA polymerase (Agilent Technologies, Santa Clara, CA), and the products were analyzed using 2% agarose gels. The following primers were used for PCR:
*il1b* Fw: 5'- GCAGAGGAACTTAACCAGCT −3'
*il1b* Rv: 5'- TGCCGGTCTCCTTCCTGA −3'
*actb1* Fw: 5'- ATGGATGAGGAAATCGCTGCCCTGGTCGTTGACAA −3'
*actb1* Rv: 5'- AGAGAGAGCACAGCCTGGATGGCCACATACATGGC −3'

## Whole-mount ISH analysis

Whole-mount ISH analysis was performed as described previously (*Hasegawa et al., 2015*), except that 5% polyvinyl alcohol was included in the buffer during the color reaction. After detection of the

ISH signal, the samples were fixed with 4% paraformaldehyde (PFA) in phosphate-buffered saline (PBS) for color preservation, equilibrated with 80% glycerol, and then mounted on glass slides and photographed. The antisense *junba*, *junbb*, *fn1b*, *apoeb*, *mpx,* and *lcp1* (*l-plastin*) probes used here were described previously (*Yoshinari et al., 2009*; *Hall et al., 2012*). The *egfp* probe was synthesized from the construct pCS2-egfp, which contained the full-length *egfp* coding sequence. The *il1b* probe was directly synthesized from the *il1b* PCR product using the T7 promoter (underlined in the primer sequence) (*Thisse and Thisse, 2008*).

*il1b* probe Fw: 5'-ATGGCATGCGGGCAATATGA-3'
*il1b* probe Rv: 5'-<u>TAATACGACTCACTATAGGG</u>CTAGATGCGCACTTTATCCT-3'

For preparing tissue sections, samples were incubated in 20% sucrose in PBS for overnight at 4°C, embedded in Tissue-Tek compound (Miles) and 16 μm sections were obtained using a cryostat.

## LPS injection

*S. typhosa* LPS (Sigma–Aldrich, St. Louis, MO) was dissolved in water at 10 mg/ml and stored in aliquots at −30°C. A solution containing LPS (5 mg/ml) and phenol red (0.5%) was injected at 0.5 nl per larva into the pericardial cavity of anesthetized larvae at 2 dpf.

## Generation of Tg lines

The construct for generating Tg(*hsp70l:mCherry-2a-il1b*) was prepared by replacing the *creERt2* sequence of pT2(*hsp70l:mCherry-2a-creERt2*) (*Yoshinari et al., 2012*) with the *il1b* cassette. The *il1b* cassette was prepared by PCR-amplifying the *il1b* sequence a.a. 125–273. The signal sequence of the human IL-1β receptor antagonist (*Wingren et al., 1996*) and the HA tag were fused at the N-terminus and C-terminus, respectively (underlined in the primer sequences).

Fw *il1b* primer: 5'-GCTAGC<u>ATGGAAATCTGCAGAGGCCTCCGCAGTCACCTAATCACTCTCC TCCTCTTCCTGTTCCATTCAGAGACGATCTGC</u>AAAAACGTCTTGCAATGCACGATTTGCG-3'

Rv *il1b* primer: 5'-AGATCTCTA<u>AGCGTAGTCTGGGACGTCGTATGGGTA</u>GATGCGCACTTTATCC TGCAGCTCGAAG-3'

The construct was injected into fertilized eggs. To identify germ-line transmission, the F0 fish were crossed with each other or with WT, and the embryos produced were screened for mCherry fluorescence after heat shock.

To generate Tg(*il1b:egfp*), the *iTol2* cassette was first introduced into the BAC clone CH211-147H23 using BAC recombineering (*Suster et al., 2011*). Next, the *egfp-nitroreductase* cassette (*Grohmann et al., 2009*) was introduced into the site of the initiation codon of *il1b*. Lastly, a mixture of purified BAC DNA (125 ng/μl) and the transposase mRNA (25 ng/μl) was injected into 1-cell-stage zebrafish embryos (1 nl/embryo). The Tg was screened for EGFP expression in the lens, which is driven by the *crystalline alpha* promoter, and further tested for EGFP induction in response to fin fold amputation.

## Whole-mount immunohistochemistry

Zebrafish larvae were fixed with 4% PFA in PBS for 2 hr at room temperature (RT) or overnight at 4°C, washed thrice with PBS containing 0.1% Triton X-100 (PBTx), dehydrated with methanol, and stored at −30°C. Samples were rehydrated with PBTx and then incubated (overnight, 4°C) with anti-E-cadherin (1:1000; BD Biosciences, Franklin Lakes, NJ; RRID:AB_397580) and anti-EGFP (1:1000; Nacalai Tesque, Kyoto, Japan; RRID:AB_10013361) antibodies in blocking buffer (5% serum and 0.2% bovine serum albumin in PBTx). After extensive washing with PBTx at RT, the samples were incubated (overnight, 4°C) with anti-mouse Alexa Fluor 568 and anti-rat Alexa Fluor 488 antibodies (both 1:1000; Invitrogen). After washing with PBTx, the tail regions were isolated and mounted in 80% glycerol containing 2.5% 1,4-diazabicyclo[2.2.2]octane (Nacalai Tesque) as an anti-fading reagent. Fluorescence images were acquired using a confocal microscope (FV-1000, Olympus, Tokyo, Japan).

## Time-lapse analysis of Tg(*il1b:egfp*)

Zebrafish larvae of the BAC Tg(*il1b:egfp*) line at 1 hpa were placed on a 2% agarose-gel stage in egg water and embedded in 0.7% low-melting agarose. Time-lapse images were acquired once

every 15 min using the confocal microscope equipped with a 20× water-immersion objective lens; the acquired images were z-stacks containing 20 optical slices.

## TUNEL staining

Larvae were fixed with 4% PFA for 2 hr at RT or overnight at 4℃, dehydrated with methanol, and stored at −30℃. Apoptosis was examined using an in situ apoptosis detection kit (Roche, Basel, Switzerland). Briefly, samples were rehydrated with PBTx, treated with 10 μg/ml Proteinase K in PBTx (5 min, RT), washed with PBTx, and refixed with 4% PFA in PBS (20 min). The samples were further incubated (15 min, on ice) in a freshly prepared 0.1% sodium citrate buffer containing 0.1% Triton X-100, washed with PBTx, and reacted with the TUNEL reaction mixture at 37℃ for 1.5 hr. The reaction was terminated by washing with PBTx. The samples were mounted in 80% glycerol and fluorescence images were acquired using confocal microscopy. The TUNEL-positive cells were quantified by counting the number of cells in the area posterior to the notochords.

## Chemical treatment

Dex (D-2915; Sigma) was dissolved in dimethyl sulfoxide (DMSO) at 100 mM and stored at −30℃. The Dex solution was diluted to 100 μM with egg water and administered to zebrafish larvae at least 1 hr before fin fold amputation.

## Sudan Black B staining

Larvae were fixed with 4% PFA for 1 hr at RT, rinsed with PBS, and incubated in 0.03% Sudan black B (Sigma). After extensive washing with 70% ethanol and rehydration with PBS containing 0.1% Tween 20 (PBT), samples were mounted in 80% glycerol.

## BrdU staining

Proliferating cells were labeled with 5-bromo-2-deoxyuridine (BrdU, 5 mM) during 0–12 hpa. The labeled larvae were fixed with 4% PFA for 2 hr at RT or overnight at 4℃, dehydrated with methanol, and stored at −30℃. Immunochemical detection was performed as described (*Yoshinari et al., 2009*) and BrdU-positive cells were quantified from the acquired confocal images. Similar to the quantification of TUNEL-positive cells, the BrdU-positive cells were quantified by counting the number of cells in the area posterior to the notochords.

## Replicates

Most assessments of phenotypes and expression patterns were replicated in at least two independent experiments with comparable results. Larvae were collected from independent crosses, and experimental processing (injection, heat shock, and/or staining) was carried out on independent occasions. Exceptions to this include data presented in *Figure 1E*, *Figure 7L*, *Figure 2—figure supplement 1*, and *Figure 6—figure supplement 1B–D*. In each of these cases, multiple larvae were processed, and the obtained phenotypes were the same in all or most cases. The n is reported within the respective figures.

## Statistical analysis

Data are presented as means ± SEM. Statistical analyses were performed using Microsoft Excel 2013. For normally distributed data, differences were analyzed using Student's *t* tests; $p < 0.05$ was considered to be statistically significant.

## Acknowledgements

We thank members of the Kawakami lab. This work was supported by grants from the Koyanagi Foundation and a Grant-in-Aid for Scientific Research (C) to A Kawakami, by the Marsden Fund grant from the Royal Society of New Zealand to C. J Hall, and by the NIG-JOINT grant from the National Institute of Genetics to A Kawakami and K Kawakami, and by the National BioResource Project from Japan Agency for Medical Research and Development (AMED) to K Kawakami. TH was supported by a fellowship from the Education Academy of Computational Life Science (ACLS) at Tokyo Institute of Technology.

# Additional information

## Funding

| Funder | Grant reference number | Author |
|---|---|---|
| Japan Society for the Promotion of Science | Grant-in-Aid for Scientific Research (C) | Atsushi Kawakami |
| Japan Agency for Medical Research and Development | National BioResource Project | Koichi Kawakami |
| Royal Society of New Zealand | Marsden Fund | Christopher J Hall |

The funders had no role in study design, data collection and interpretation, or the decision to submit the work for publication.

## Author contributions

TH, Conceptualization, Formal analysis, Investigation, Writing—original draft, Writing—review and editing; CJH, Resources, Data curation, Supervision, Funding acquisition, Investigation, Methodology; PSC, Data curation, Supervision, Funding acquisition; GA, Resources, Investigation, Methodology; KK, Resources, Funding acquisition, Methodology; AKu, Supervision, Project administration; AKa, Conceptualization, Resources, Data curation, Formal analysis, Supervision, Funding acquisition, Investigation, Methodology, Writing—original draft, Project administration, Writing—review and editing

## Author ORCIDs

Koichi Kawakami, http://orcid.org/0000-0001-9993-1435
Atsushi Kawakami, http://orcid.org/0000-0001-9461-6372

## Ethics

Animal experimentation: This study was performed in strict accordance with the recommendations in the Act on Welfare and Management of Animals in Japan and the Guide for the Care and Use of Laboratory Animals of the National Institutes of Health. All of the animals were handled according to the Animal Research Guidelines at Tokyo Institute of Technology. The protocol was approved by the Committee on the Ethics of Animal Experiments of the Tokyo Institute of Technology. All surgery was performed under tricaine (3-aminobenzoic acid ethyl ester) anesthesia, and every effort was made to minimize suffering.

# Additional files

## Supplementary files

• Supplementary file 1. The list of transcripts that are upregulated or downregulated in the amputated fin fold of the *clo* mutant. The table of upregulated genes shows the list of transcripts whose final RPKMs of the amputated *clo* mutant at 6 hpa are more than two times than those of the amputated WT. The table of downregulated genes shows the list of top 100 transcripts whose final RPKMs of the amputated *clo* mutant at 6 hpa are downregulated compared with those of amputated WT. Transcripts with low expression in the *clo* mutant (final RPKM <4) were excluded. Final RPKM, the reads per kilobase of exon model per million mapped reads. Transcripts that are shown in *Figure 1A* are highlighted.

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
