## [Decision Letter]

Thank you for submitting your article "Transient inflammatory response mediated by interleukin-1β is required for proper regeneration in zebrafish fin fold" for consideration by *eLife*. Your article has been favorably evaluated by Robb Krumlauf (Senior Editor) and three reviewers, one of whom is a member of our Board of Reviewing Editors. The reviewers have opted to remain anonymous.

The reviewers have discussed the reviews with one another and the Reviewing Editor has drafted this decision to help you prepare a revised submission

Summary:

This interesting paper from the Kawakami lab follows up on their earlier publication reporting that fin regeneration fails to occur in zebrafish cloche mutants due to the lack of myeloid cells.

In this new study, they show that *il1b* is a likely factor involved in the lack of fin regeneration in cloche mutants using both loss- and gain-of-function analyses.

Interestingly, *il1b* appears to be expressed by epithelial cells, not myeloid cells, and macrophages appear to be necessary for the resolution of inflammation/end of *il1b* expression. Prolonged *il1b* expression leads to apoptosis.

Essential revisions:

1) When do macrophages arrive near the wound, and how does that relate with the down-regulation of *il1b* expression? Defining macrophage as a key cell type mediating cell survival during fin fold regeneration is one of the major conclusions of this paper. Thus, the characterization of macrophages in this mechanism would need to be elaborated more. Is there any evidence that macrophage infiltration in the vicinity of *il1b*-positive epithelial cells occurs after the initial priming stage (Figure 8)? Is it possible that they infiltrate immediately after the injury and change their function afterwards to suppress *il1b* signals? Or, is it possible that macrophages are not residing in the regenerating area and mediate the suppressive function remotely?

These questions could be addressed by using mpeg1 reporter fish, its reporter construct injection, or in situ hybridization.

2) Overexpression of *il1b* induced pro-regenerative gene expression in uninjured fins (Figure 7). This result may also be interpreted as an indirect regenerative response to cell death induced in uninured fins with excessive *il1b* (Figure 6—figure supplement 1), not as evidence for the direct role of *il1b* in regeneration priming.

One way to address this question is by doing TUNEL staining at a time point when the regenerative genes are induced in the uninjured, *il1b*-overexpressing fin to determine whether these genes are expressed unrelated to or earlier than Il1b-mediated cell death induction.

---

## [Author Response]

*Essential revisions:*

*1) When do macrophages arrive near the wound, and how does that relate with the down-regulation of il1b expression? Defining macrophage as a key cell type mediating cell survival during fin fold regeneration is one of the major conclusions of this paper. Thus, the characterization of macrophages in this mechanism would need to be elaborated more. Is there any evidence that macrophage infiltration in the vicinity of il1b-positive epithelial cells occurs after the initial priming stage (Figure 8)? Is it possible that they infiltrate immediately after the injury and change their function afterwards to suppress il1b signals? Or, is it possible that macrophages are not residing in the regenerating area and mediate the suppressive function remotely?*

*These questions could be addressed by using mpeg1 reporter fish, its reporter construct injection, or in situ hybridization.*

Migration of myeloid cells in response to tissue injury has been described in preceding studies using transgenic zebrafish and ISH analysis (Li et al., 2012; Yohisnari et al., 2009). According to the reviewer’s suggestion, we further examined temporal changes of neutrophil and macrophage accumulation, respectively, in the injured fin fold. We observed that both of neutrophil and macrophage increased until 3 hpa, reached at a maximum number between 3-6 hpa in the vicinity of *il1b*-expressing site, and then decreased thereafter. The timing coincides with the stage when the initial *il1b* expression in WT was downregulated, suggesting a possibility that myeloid cell migration could play a role for attenuating *il1b* expression. However, we cannot conclude that it is necessary for attenuating *il1b* expression. Because, we have previously shown that an anti-apoptosis activity exists in the WT body fluid irrespective of tissue injury (Hasegawa et al., 2015).

Figure 4: We added the ISH analyses of neutrophil and macrophage accumulation in amputated fin fold.

Subsection “Macrophages support the survival of regenerative cells”, first paragraph: We added a description of myeloid cell accumulation in amputated fin fold in the Results section.

Discussion, fifth paragraph: We also added a discussion regarding the significance of myeloid cell migration to injured site.

Figure 8: The illustration in Figure 8 was revised, because we cannot conclude that macrophage infiltration in the vicinity of *il1b*-positive epithelial cells is necessary for attenuating *il1b* expression.

*2) Overexpression of il1b induced pro-regenerative gene expression in uninjured fins (Figure 7). This result may also be interpreted as an indirect regenerative response to cell death induced in uninured fins with excessive il1b (Figure 6—figure supplement 1), not as evidence for the direct role of il1b in regeneration priming.*

*One way to address this question is by doing TUNEL staining at a time point when the regenerative genes are induced in the uninjured, il1b-overexpressing fin to determine whether these genes are expressed unrelated to or earlier than il1b-mediated cell death induction.*

In the used Tg(*hsp70l:il1b*), multiple heat shock induction (5 times taking 2 days) was required before detecting the aberrant apoptosis in uncut WT fin fold. This is probably because the macrophages in WT suppress the inflammatory effect of *Il1b* overexpression. The induction of regenerative genes shown in Figure 7 was observed at 6 hpa, a stage before the apoptosis by *il1b* overexpression was evident. Indeed, we confirmed that apoptosis was not detected in the uncut fin fold after two HS inductions within 2 days.

Figure 7—figure supplement 1: We added new data.

Subsection “Normal inflammation mediated by Il1b is required for fin fold regeneration”, last paragraph: We added sentences describing that the expression of the regeneration-induced genes was not due to an indirect regenerative response to cell death.